# Adaptive Regression Prefetching Algorithm by Using Big Data Application Characteristics

Mengzhao Zhang [1], Qian Tang [1], Jeong-Geun Kim [2], Bernd Burgstaller [1] and Shin-Dug Kim [1,*]

[1] Department of Computer Science, Yonsei University, 50 Yonsei-ro, Seodaemun-gu, Seoul 03722, Republic of Korea

[2] School of Computer Science and Engineering, Kyungpook National University, 80 Daehak-ro, Buk-gu, Daegu 41566, Republic of Korea

* Correspondence: sdkim@yonsei.ac.kr

**Abstract:** This paper presents an innovative prefetching algorithm for a hybrid main memory structure, which consists of DRAM and phase-change memory. To enhance the efficiency of hybrid memory hardware in serving big data technologies, the proposed design employs an application-adaptive algorithm based on big data execution characteristics. Specifically optimized for graph-processing applications, which exhibit complex and irregular memory access patterns, a dual prefetching scheme is proposed. This scheme comprises a fast-response model with low-cost algorithms for regular memory access patterns and an intelligent model based on an adaptive Gaussian-kernel-based machine-learning prefetch engine. The intelligent model can acquire knowledge from real-time data samples, capturing distinct memory access patterns via an adaptive Gaussian-kernel-based regression algorithm. These methods allow the model to self-adjust its hyperparameters at runtime, facilitating the implementation of locally weighted regression (LWR) for the Gaussian process of irregular access patterns. In addition, we introduced an efficient hybrid main memory architecture that integrates two different kinds of memory technologies, including DRAM and PCM, providing cost and energy efficiency over a DRAM-only memory structure. Based on the simulation-based experimental results, our proposed model achieved performance enhancement of 57% compared to the conventional DRAM model and of approximately 12% compared to existing prefetcher-based models.

**Keywords:** graph processing; regression algorithm; data prefetching; hybrid main memory; memory management

## 1. Introduction

The utilization of big data processing has become a popular trend in recent years [1,2]. As big data applications continue to advance and become more intricate, the interest in big data has grown and resulted in increasingly complex graph structures. This complexity has caused irregular and complicated memory access patterns in graph processing, creating a significant bottleneck [3]. To overcome this, prefetchers are employed to predict when to fetch any particular type of data into the cache memory to reduce memory access latency. Such prefetchers have demonstrated substantial improvement in mitigating the miss penalty for conventional memory access patterns, e.g., with stream prefetchers [4], stride prefetchers [5], and a global history buffer (GHB) [6]. However, conventional pattern-recognition-based prefetching approaches are inappropriate to identify mixed and irregular memory access streams such as in-direct memory access and pointer-chasing memory accesses. Therefore, the key to achieving accurate and effective prefetching is to exclude inapplicable memory access patterns based on lightweight and learning-based prefetching mechanisms. By analyzing the graph data structure and access patterns, this paper proposes a deformed regression prefetching scheme that uses an invalid address index to effectively hide the cache invalidation delay and improve the overall performance.

Many studies have been carried out on new hybrid memory technologies for non-volatile memory (NVM). PCM has the advantages of low power consumption, high density, low latency, and byte-addressability. However, it has disadvantages of its own: writing power consumption is higher than for reading, write latency is higher than reading, and write counts of PCM are limited [7]. The asymmetry problem regarding read and write in PCM should receive more attention when using hybrid PCM and DRAM memory [8]. This work proposes a new type of efficient memory page management mechanism called the hybrid main memory management unit (HMMMU) based on DRAM-PCM hybrid memory architecture. The main idea of this mechanism is to allocate the data pages into the appropriate memory space. For example, this mechanism can put data with a high access potential into the high-performance zone in the main memory to reduce migration operations and then improve the system performance.

Therefore, the core problem is how to distinguish memory access patterns and place the most frequent data on DRAM. In response to this problem, a prefetch table is designed that can monitor memory requests and analyze access patterns in real time and, based on the time and space limitations of the memory access sequence, an arbitrator is designed to manage three different types of prefetch engines. In particular, for irregular memory access patterns, the optimized linear regression algorithm is used, and partial lines are added to increase the weight, thereby improving the prefetch efficiency.

To quantitatively evaluate the proposed DRAM-PCM hybrid main memory system, memory requests from applications will be logged in trace files by using Pin Tool, version 3.0 [9]. The simulator can access multiple trace files and perform multiple fetches simultaneously. A detailed description of the workload is given in Section 4. The workload is executed on the dataset generated by the data generator called LDBC [10]. Our simulation-based evaluation shows that our proposed model, DRAM-PCM hybrid memory architecture with an optimized regression prefetching method, demonstrated a significant improvement in execution time and energy consumption. Specifically, our proposed model achieved significant improvement over the conventional DRAM-only main memory architecture model, with a 56% reduction in execution time and a 57% decrease in energy consumption. Our proposed model achieved a reduction in total execution time by 8% and power consumption by 13% when compared to existing models such as GHB, access map pattern matching (AMPM), spatial storage stream (SMS), best-offset, and DynamicR. Furthermore, it also improved the total hit rate by 2% on average.

The prefetch technology is essentially a memory access sequence-prediction technology, and the prefetch strategy should consider the memory access behavior of the application [11]. This section will analyze and summarize the patterns of the address sequence when the cache is missed, analyze the miss ratio in the cache, and design the prefetch strategy based on the analysis results. Figure 1 shows a comparison of the miss rates of various access patterns in the test program. As shown in Figure 1, the miss rate of irregular accesses is much larger than that of the regular access patterns. At the same time, the delay in accessing external memory caused by the misses of irregular access patterns is one of the main bottlenecks restricting the improvement of overall performance.

Therefore, improving the performance of irregular prefetch patterns has a greater effect on improving the performance. To prevent performance degradation by irregular memory access streams from graph-processing workloads [12], we propose a DRAM buffer-based hybrid main memory architecture with a prefetching method based on an optimized locally weighted regression (LWR) algorithm. To address the challenges of traditional DRAM-based main memory limitation, increasing energy consumption, and the high-cost issue, our proposed solution introduces the DRAM-PCM-based hybrid main memory architecture based on our novel data management method. The design in this paper is a universal solution for all DRAM-based main memory architectures. Because the proposed model has the advantage of aggregating hot data, it can well serve the hybrid memory system that consists of non-volatile memory (NVM) with a relatively limited read-and-write lifetime. It

can significantly increase the lifetime of non-volatile memory while reducing the memory miss rate.

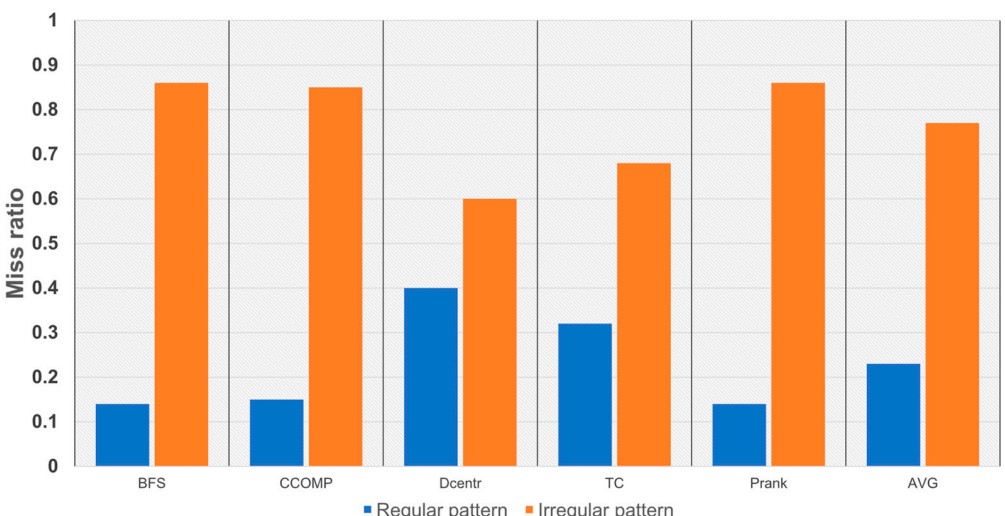

**Figure 1.** Miss ratio from different patterns for graph- processing workloads.

This paper makes the following contributions:

1. We conducted an analysis of complicated and indirect memory access patterns from graph workloads to design a data prefetch algorithm by determining dynamic memory access patterns as a design application method utilizing big-data execution characteristics.
2. We designed an adaptive optimized regression prefetch scheme that can select from a dynamic set of prefetch engines by using a machine-learning approach.
3. We proposed a novel page-management mechanism that leverages the cost-effectiveness of PCM's characteristics by efficiently utilizing both DRAM and PCM.

The subsequent sections of this paper are structured as follows: Section 2 provides an overview of the related research on prefetching methods and hybrid main memory architectures, while Section 3 presents our proposed DRAM-PCM hybrid memory model with the optimized linear regression algorithm. Section 4 conducts a comparative analysis of the proposed system with existing models. In Section 5, we present the experimental results obtained from our in-house simulation environments with various hybrid memory models.

## 2. Related Work

### 2.1. Prefetching Scheme

Hardware prefetching utilizes hardware dynamics to analyze the execution process of the program and prefetches according to the memory access history. Although the complexity of hardware design has increased, the efficiency is higher.

Norman P. Jouppi et al. [4] introduced stream buffer prefetching technology, prefetching the subsequent addresses of the cache invalidation address at the memory access invalidation module level and putting the prefetched data into the storage unit of the FIFO structure, that is, the stream buffer.

Fu et al. [5] proposed a stride prediction-based prefetching mechanism to enhance the performance for matrix-based calculations. It was implemented using a stride prediction table (SPT) to calculate the stride distances. The stride prefetcher obtains the difference between the address currently accessed by the processor and the previously generated address value and stride, and then adds the difference to the accessing address to perform the next prefetch.

Nesbit et al. [7] proposed a prefetching mechanism that employs an index table in conjunction with a global history buffer (GHB), and the delta correlation prefetching scheme

that tracks cache events and memory requests information to find appropriate address deltas for generating the next prefetching candidates.

Somogyi et al. [11] proposed an SMS prefetcher for spatial relations, which uses context data (PC and region offset) to identify spatial patterns that are not necessarily continuous. SMS focuses on finding spatial patterns using two tables, the accumulation table and the filter table. SMS trains spatial patterns by tracking demand requests across a spatial region as page granularity. SMS records detailed information of demand requests, including values of the program counter (PC) and spatial bit-patterns, and tries to predict spatial patterns with this history to generate the next prefetching candidates.

Ishii et al. [13] proposed a model that manages a memory access map to store the layout of spatial memory accessing records called the AMPM prefetcher. By analyzing more than two strides of the request access pattern, the prefetcher can predict the closest block for the next request.

Michaud et al. [14] designed a best-offset prefetcher that builds upon the sandbox prefetcher and attempts to enhance its timeliness. The best-offset prefetcher depends on the sandbox prefetcher that chooses the best single offset to fit current memory access streams. It evaluates various offsets to predict the offsets that are expected to produce timely prefetches.

To consider prefetch timeliness, Yun et al. [15] introduced a simple linear regression preprocessing technique. This involves first finding frequently used hot data, and then, second, using simple linear regression to obtain the next data that need to be prefetched and stored in the buffer.

Based on the design of Yun et al. [15], Kim et al. [16] devised a multi-algorithm composite prefetching method that employs history tables-based approaches, by comprehensively analyzing complicated memory request patterns. The method includes an efficient next-line prefetch engine and a linear regression engine.

Building upon the excellent work of Yun and Kim et al., we published a more advanced dynamic recognition prefetch scheme in our previous research [17]. The algorithm can automatically switch the required prefetch algorithm according to the memory access patterns. Then it performs a low-cost polynomial fitting through pre-training, which makes up for the limitation of the linear regression algorithm. Finally, according to the discrete characteristics of the data, the algorithm can dynamically adjust the number of prefetch pages to ensure the accuracy and efficiency of the prefetch engine.

With the development of memory management unit (MMU) technology and the need to further reduce the memory miss rate, prefetching is becoming more advanced and complex. Although conventional linear prediction has a good performance in the graph processing system, there are still many memory accesses that do not conform to the linear characteristics. Covering those accesses requires advances in the efficiency of prefetching. However, with the increasing complexity, the requirements for hardware devices gradually increase, e.g., with the polynomial fitting engine in [17]. Even if simplified algorithms are used with the objective of reducing complexity, there will still be over-fitting phenomena in the experiment. To enhance the robustness and reduce the system burden while ensuring prefetching efficiency, we propose a new type of efficient memory management mechanism and a more efficient prefetching algorithm in this paper.

### 2.2. Hybrid Memory

Recently, many studies have been carried out on hybrid main memory combined with conventional DRAM, which utilizes the advantages of NVM.

Choi et al. [18] proposed a method using an NVM-capacity management policy for a hybrid memory system to minimize the number of write operations and guarantee the lifetime of NVM cells and memory system performance with the dynamic way-selection algorithm. In addition, a method to disable NVM was mentioned in this work to prevent fetching new blocks into inactivated NVM devices.

Ramos et al. [19] proposed a multi-queue-based page placement method that operates page migration between DRAM and PCM devices for the page-granularity-based hybrid main memory system. In addition, the memory controller features a ranking-based page placement scheme that efficiently prioritizes pages based on their popularity and write intensity. This approach employs the page migration of top-ranked pages to DRAM.

## 3. Main Architecture

### 3.1. Overall Architecture

The proposed model, as shown in Figure 2, is comprised of the following six modules: namely, DRAM buffers that store prefetch addresses, arbitrators responsible for managing the prefetch engine, and prefetch tables that record recently accessed addresses from the main memory and the DRAM-PCM hybrid main memory. Additionally, the model incorporates different prefetch mechanisms for various memory access patterns.

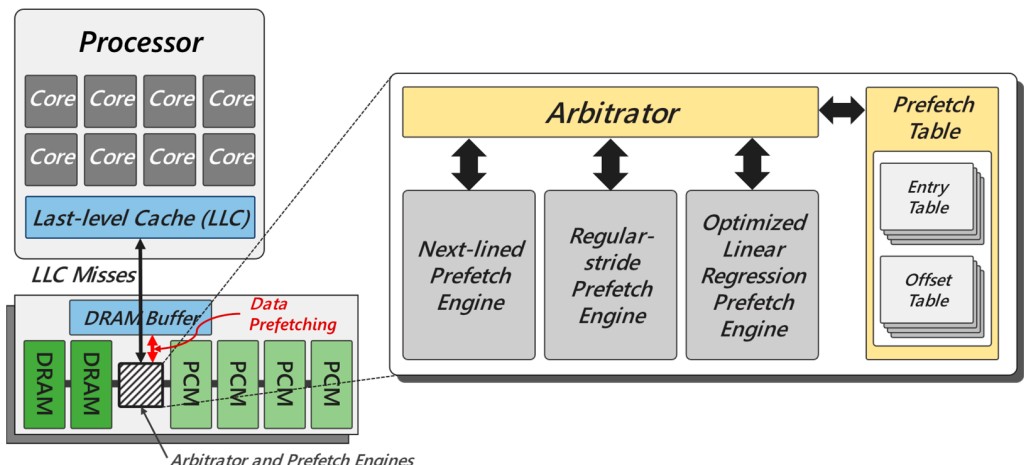

**Figure 2.** Overall architecture.

### 3.1.1. DRAM Buffer

As shown in Figure 2, the DRAM buffer provides a buffer region for storing the addresses generated by the prefetch engines. When an LLC miss occurs and there are no addresses in the DRAM buffer, the hybrid main memory system is accessed.

### 3.1.2. Hybrid Main Memory Management Unit

We designed a DRAM prefetch buffer that is placed between the last-level cache (LLC) and the main memory layer, which serves to store prefetched data. In addition, our proposed model incorporates a hybrid main memory system that includes a limited amount of DRAM device and a larger background memory area composed of PCM device. Moreover, our model incorporates a hybrid main memory management unit that efficiently manages these distinct types of memories. Specifically, the hybrid main memory, which includes the prefetching buffers, operates using a flat physical memory address scheme with an LRU-based page replacement policy.

### 3.1.3. Arbitrator

The operation flow of the arbitrator is demonstrated in Figure 3. As shown in the figure, the arbiter is capable of determining whether or not to update the prefetch table, but also of acting on the prefetch engine to intelligently identify the memory-access mode, and then select the most suitable prefetch engine to perform prefetching.

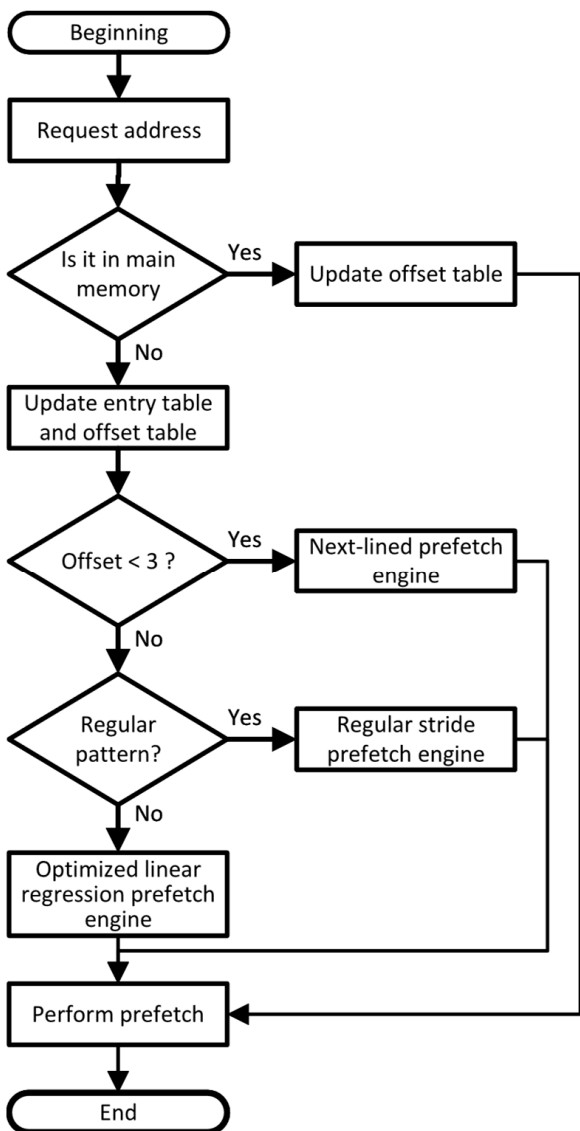

**Figure 3.** Operational flow of Arbitrator.

*3.2. Cache Miss Analysis*

Through the analysis of the program access mode, the access mode from the perspective of cache can be summarized into three patterns: a stable stride memory access pattern, a linear transformation stride pattern, and an irregular memory access pattern, as shown in Table 1.

**Table 1.** Cache Miss Address Patterns.

| Address Sequence Pattern | Address Sequence Instance |
| --- | --- |
| Stabile stride memory access pattern | 0x80000080, 0x80000080, 0x80000100, 0x80000180 |
| Linear transformation stride pattern | 0x80000000, 0x80000080, 0x80000180, 0x80000300 |
| Irregular memory access pattern | Other |

*3.3. Regression Analysis*

Linear regression is a machine-learning method that models the tendency between a scalar response and multiple explanatory variables [20]. The following equation shows the formula for the regression hypothesis.

$$y = \theta_0 + \theta_1 x_1 + \theta_2 x_2 + \ldots + \theta_n x_n \tag{1}$$

As illustrated in Equation (1), this technique is used to solve a model by minimizing the mean square error, commonly referred to as the least square method, which finds the best-fitted line that minimizes the sum of all distances from each sample.

*3.4. Prefetch Table*

The traditional prefetch mechanism mostly uses the memory access instruction address as the index of the query prefetch table. To avoid increases in the complexity of the hardware design caused by the increase in the data path, it also makes the logic design more direct. This study uses the memory access address as the query prefetch table and obtains the index of the table entry. The key to prefetching technology lies in the prediction of memory access addresses. Therefore, prefetching technology is another application of value prediction technology. In the prefetch mechanism introduced in this study, the concept of offset is introduced to increase the efficiency of cache prefetching.

Similarly, the entry table and offset table are both used in the prefetch mechanism of the new prefetch table. The entry address in the entry table represents the physical address of the entry bit (46 bits). In all 64 bits, the prefetch number represents the prefetch times of each entry, and the hit represents the prefetch success. The offset table includes 256 offsets with 64 entries. The structure of the prefetch table is shown in Figure 4.

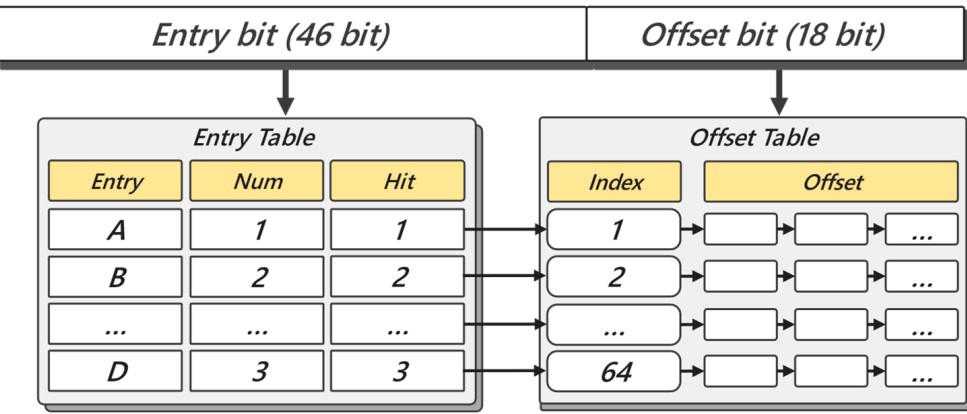

**Figure 4.** Prefetch table schematic diagram.

3.4.1. Entry Table

The update of the entry is controlled by the arbitrator and this table is updated using the prefetch controller, as shown in Figure 5. When the LLC requests data to be transferred to the main memory and a miss occurs, the arbitrator updates the entry table as follows: If the missing data is the same as in the entry table, only the offset table needs to be updated; however, if it is different, a new entry needs to be inserted, and the offset is updated accordingly. In addition, if the entry table misses and is empty, it will be executed according to the least recently used (LRU) strategy.

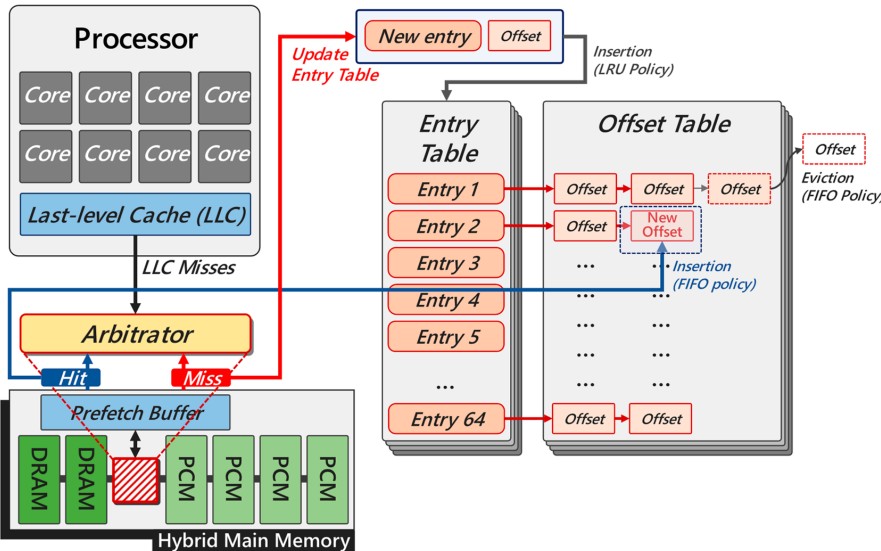

**Figure 5.** Update flow of prefetch table.

### 3.4.2. Offset Table

Unlike the selective update of the entry table, the offset table updates every request from the LLC, and the offset table can be attached only to the entry table; that is, it cannot exist alone. In addition, the replacement strategy for the offset table is first in and first out (FIFO). Given that the prefetch table can record the requested data from the LLC miss, the prefetch table can record the memory access model to predict the next prefetch data. This mechanism also greatly improves performance.

### 3.5. Prefetch Engine

Based on the complexity of the memory access patterns, this study proposes three prefetch models for different usage scenarios. The fast-response model consists of two prefetch engines: the next-lined prefetch engine and the regular-stride prefetch engine. The intelligent mode is an adaptive Gaussian-kernel-based optimized LWR prefetch engine.

### 3.5.1. Next Lined Prefetch Engine

The next-lined prefetch engine was designed based on a prefetch table. When there are fewer than three offset entries, it is not possible to determine the regular access mode. When the offset is greater than three, using the next lined prefetch reduces the prefetch rate of the regular pattern. Therefore, three was chosen as the critical point to determine whether the prefetch engine was used or not. The prefetch operation method is shown in Algorithm 1. When the offset number of the last entry requested is fewer than three, according to the principle of time and space limitations, the prefetch table does not perform pattern recognition. Instead, the next page is directly prefetched.

---

**Algorithm 1:** Next Lined Prefetching Algorithm.

---

//Step 1:
**for** offset **in** *Entry table*
    **find** offset size

//Step 2:
**if** offsetSize < 3
    Offset x ← Last offset x
    *PrefetchAdd* ← 0
**else**
    *PrefetchAdd* ← Nextline **Prefetch**(reqAdd)
**return PrefetchAdd**

---

### 3.5.2. Regular Stride Prefetch Engine

The regular stride prefetch engine also performs pattern recognition based on the prefetch table. The prefetch operation method is shown in Algorithm 2.

When the offset number of the last entry requested is equal to three, the delta is obtained. When the last two consecutive deltas are equal, the prefetch engine will perform the next prefetch according to the increase in the delta.

Once the most recent memory request offset surpasses three offsets within the entry, the recognition phase for the memory access pattern is triggered, and it utilizes the last four accessed offsets in the entry of the memory address that caused the phase. If the three deltas in the last four consecutive memory accesses are identical, the previous prefetch strategy is used to generate a prefetching candidate. Otherwise, the interval will be obtained; if two of the three deltas are the same within the interval, the prefetching interval will be increased.

---

**Algorithm 2:** Regular stride prefetch engine.

---

//Step 1:
**for** offset **in** *Offset table*
      **find** last offset size

//Step 2:
**if** offsetSize == 3
      Delta[i] = offset[x-i] − offset[x-i-1]
      **if** Delta [0] = Delta [1]
            *PrefetchAdd* ← reqAdd + Delta [0]

//Step 3:
**else**
      DeltaInter[i-1] = Delta[i] − Delta[i-1]
      **if** Delta [0] == Delta [1] && Delta [1] = Delta [2]
            *PrefetchAdd* ← reqAdd + Delta [0]
      **else if** DeltaInter [0] == DeltaInter [1]
            *PrefetchAdd* ← reqAdd + Delta [0] + DeltaInter [0]
**return PrefetchAdd**

---

### 3.5.3. Optimized LWR Prefetch Engine

When the prefetch table cannot be used for regular pattern recognition, the arbiter uses an irregular prefetch engine for prefetching. The irregular prefetch engine searches for frequently accessed data in the prefetch table, specifically targeting the entry with the highest number of offsets to retrieve the most commonly used data. Using linear regression to obtain an optimal regression coefficient vector $w$ enables the prediction of the value of $y$ through the corresponding expression as follows:

$$y = ax \tag{2}$$

Consequently, we obtain value $a$ through the least square method and Gaussian erasure, followed by the prediction of the next value $y$. First, we look for the one with the most offsets in the entry table, and then arrange them in order, where $x$ is the sequential address, and the y-axis is the offset value; thus, the sequences of $x$ and $y$ can be trained as a set $(x, y)$; then, the regression algorithm is used to obtain the regression coefficient F, and the next step is to prefetch the next F(n + 1). The above is a simple linear regression algorithm, but in fact, most data may not be described by a linear model, and it is possible that not only can they not fit all data points well, but that the error is also very large. To solve the problem of a non-linear model building a linear model, when the value of a point is predicted, the point close to this point was chosen instead of all points. Based on this idea, a locally weighted regression algorithm was developed, where the closer the others are to a point, the greater the weight, and the greater the contribution to the regression coefficient.

For the above algorithm, optimization operations were performed for special scenarios. When the predicted value of a certain $y$ is obtained, the regression coefficient $w$ is needed, but for the data in the sample, the closer the distance $x$ is, the larger the value is. If the distance is farther, it gives a small weight, which makes the predicted value $y'$ for $x$ more suitable for the sample data.

To choose the right weight, a higher weight of the sample point from a given $x$ is needed. LWR uses the kernel to give higher weight to the nearby points, with the most commonly used method being the Gaussian kernel function, which corresponds to the following expression:

$$\omega(x,\,i) = e^{-\frac{(x-x_i)^2}{2k^2}} \tag{3}$$

It can be seen from the formula that if the distance between $x$ and $x_i$ is smaller, $\omega$ will be larger. However, because the x-axis is arranged in the order of visits, it has no reference value; therefore, for application, $x$ is changed to $y$; that is, the new expression is the deformed Gaussian kernel function that can be expressed as:

$$\omega(y,\,i) = e^{-\frac{(y-y_i)^2}{2k^2}} \tag{4}$$

After weighting each point, the regression algorithm is executed to obtain F(n), and F(n + 1) is then prefetched. The prefetch operation method is described in Algorithm 3.

---

**Algorithm 3:** Optimized Linear Regression Engine.

---

//Step 1:
**for** address **in** *Entry table*
    **if** offsetTable[i].dataSize > offsetSize
        offsetEntries ← offsetTable[i]

//Step 2:
x ← i+1
y ← offset[i]Value
cc ← calCC(offsetEntries)
Δoffset = offsetTable.max - offsetTable.min
k ← calK(cc, Δoffset, pageSize) *//function 5*
weight[i] ← calWeight(y[i], y[maxNum], k) *//function 4*

//Step 3:
*//find regression coefficient*
sortAscendingOrder(sortArray, offsetTable[i])
y[i] ← weight[i] * y[i]
coefficient ← calRegressionCoefficient(sortArray)

*//predict offset*
predictOffsetValue(coefficient, offsetSize + 1)
*PrefetchAdd* ← entryTable[i] + predictedoffset

**return PrefetchAdd**

---

### 3.5.4. Adaptive Hyperparameter Setting for Gaussian Kernel

In weight function (4) in Algorithm 3, $y_i$ is the center of the kernel function. As a hyperparameter, $k$ controls the radial range of the radial basis function, that is, the data width. However, due to the different amounts of data in each entry, the value of $k$ may cause side effects. If the data width is too large, the drop gradient of weights will disappear in the entry with smaller offset differences. Meanwhile, if the data width is insufficient, the group with larger offset differences and more samples will have a high percentage of weights close to 0. Therefore, a fixed $k$ value cannot be applied well to all entries. It is

necessary to design an adaptive dynamic *k*-value algorithm to match all entries in different situations.

We described the design and functionality of the history table in Section 3.4. As shown in Figure 4, each entry can cover a memory address space of $2^{18}$. To determine the data width of the Gaussian kernel, we first calculate the difference between the highest offset and the lowest in the entry. Then we take the page size set by the system as the step size to calculate the preliminary data width. The amount of the address space covered by the sample is not the only factor that needs to be considered. The degree of linear correlation between the data is also a decisive factor affecting the weight distribution. The more obvious the linear features, the higher the confidence in the data acquisition weights. Therefore, we introduce the linear correlation coefficient (*cc*) as an activation function to dynamically adjust the weight assignment more finely. The detailed function is as follows:

$$k = cc \frac{(y_{max} - y_{min})}{2 pageSize} \tag{5}$$

## 4. Evaluations

This section presents the workload characteristics and simulation configurations.

### 4.1. Workload Characteristics

In this section, we conduct the performance evaluation with the GraphBig benchmark suite and five selected representative graph processing kernels (workloads) including breadth-first search (BFS), connected component (CCOMP), degree centrality (DCentr), triangle count (TC), and PageRank (Prank). The BFS algorithm is widely used to traverse a graph's vertices for various purposes in graph-based computing, whilst the CCOMP is also implemented based on the BFS traversal operation. DCentr is a graph centrality algorithm which is used to analyze social network graphs. Furthermore, the PageRank algorithm is used to enhance the accuracy of the web search engine by Google Search to rank the priority and importance of web pages.

Table 2 presents the basic usage of the graph processing workloads and the structure with memory access patterns. As shown in Table 2, irregular memory access patterns are prevalent in graph processing applications.

**Table 2.** Workloads of Graph Processing.

| Workload | Computation Type, Feature, Use Case |
|---|---|
| BFS | Graph traversing algorithm, indirect memory access patterns (e.g., irregular, read-intensive memory requests), similarity search and finding maximum flow |
| Connected component (CCOMP) | Connectivity computation for graphs, irregular memory access with read-intensiveness, social graph analysis |
| Degree centrality (Dcentr) | Connectivity computation for graphs, indirect and irregular memory accesses, social graph analysis |
| Shortest (SPath) | Finding global optima algorithm for graph structure, indirect memory accesses with read-intensive characteristics, street navigation |
| Page Rank (Prank) | Iterative computations for graph analysis, Compute-intensive with indirect memory requests, prioritizing web pages |

To gain a better understanding of memory localities, we graphically analyzed the memory access patterns of each graph processing workload based on our memory access traces. Figure 6 illustrates the graph processing based on 1 million cache misses from the last-level cache (LLC) to the main memory system. The visualization on the x-axis,

representing the data accessing sequence over time, and the y-axis, representing the memory address, shows that the access patterns of graph processing are heavily irregular, covering a wide range of memory space.

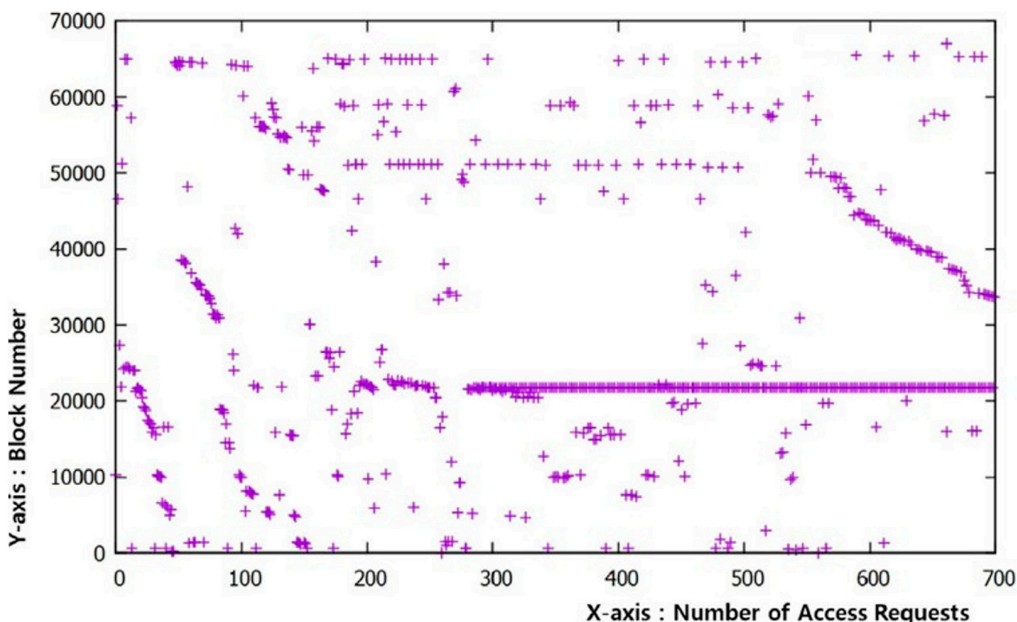

**Figure 6.** Memory access patterns of graph processing.

In addition, the program for generating benchmarks is LDBC [4], which was used to generate the graph structure data including nodes, edges, and connection information, and the vertex and edge sizes were 1 and 28.82 MB each for generating data.

*4.2. Simulation Configurations*

The proposed model was evaluated by our in-house trace-driven memory system simulator, which needed memory traces. Hence, we employed Pin-3.0 [8] to gather memory traces by running $\times$86 applications directly upon our host environment. Moreover, the generated trace is also suitable for measuring multicore-based environments. Table 3 presents the configuration details of our proposed model. The simulator used in our experiments was equipped with four cores, each running at a frequency clock of 4.0 GHz. The private L1 instruction and data caches had a capacity of 32 KB per core, and were organized as an 8-way set associativity structure with a 64-byte cache-line size. The private L2 unified caches had a capacity of 256 KB per core, and were organized as a 4-way set associativity with a 64-byte block size. Finally, the shared L3 unified caches had a capacity of 8MB per core, and were organized in a 16-way set associativity with a 64-byte block size.

Our proposed hybrid main memory model is comprised of a small DRAM module (128 MB) and a larger PCM module (2 GB), both based on fully associative organization with a 4 KB page granularity. Both memories adopt an LRU page replacement policy to swap pages between DRAM and PCM devices. A prefetch buffer is present for prefetching data from the hybrid main memory to the last-level cache layer, with the DRAM buffer having a capacity of 16 MB to store prefetching addresses generated by prefetch engines.

**Table 3.** Graph Processing Workloads.

| Processor | Quad-Cores, 4 GHz |
|---|---|
| L1 Instruction Cache (per core, private) | 32 KB, 8-way set associativity, 64-byte cache line size, LRU replacement |
| L1 Data Cache (per core, private) | 32 KB, 8-way set associativity, 64-byte cache line size, LRU replacement |
| L2 Unified Cache (per core, private) | 256 KB, 4-way set associativity, 64-byte cache line size, LRU replacement |
| L3 Cache (LLC) (per processor, shared) | 8 MB, 16-way set associativity, 64-byte cache line size, LRU replacement |
| DRAM Buffer | 16 MB, fully associative, 4 KB page size (managed as page-granularity), LRU replacement |
| DRAM | 128 MB, fully associative, 4 KB page size (managed as pagegranularity), LRU replacement |
| PCM | 2 GB, fully associative, 4 KB page size (managed as page-granularity), LRU replacement |

The prefetch table consists of an entry table and an offset table, with 64 entries in the former, each containing an entry memory address, a prefetch address value, and a prefetch hit count. The entry table has a capacity of 6.4 KB, with each entry having a capacity of 100 bits (46 bits + 32 bits + 32 bits). The offset table comprises 256 entries, each with an 18-bit offset bit, and has a capacity of approximately 4.6 KB, resulting in a total prefetch table size of around 11 KB. The entry table uses the LRU replacement policy, while the FIFO replacement policy is employed for the offset table. Table 4 outlines the simulation parameters employed to assess the efficacy of our prefetcher-based model compared to other prefetcher-based hybrid main memory architectures [21,22].

**Table 4.** Simulation Parameters.

| Parameter | DRAM | PCM | HDD |
|---|---|---|---|
| Write latency | 20–50 ns | 1 ns | 5 ms |
| Read latency | 20–50 ns | 50 ns | 5 ms |
| Write energy | 1.2 J/GB | 6 J/GB | 65 J/GB |
| Read energy | 0.8 J/GB | 1 J/GB | 65 J/GB |
| Idle power | 100 mW/GB | 1 mW/GB | 10 W/TB |
| Density | 1× | 4× | N/A |
| Cost | 4× | 1× | N/A |

### 4.3. Performance Evaluation

To measure the performance of the proposed model, some analyses on the impact of the offset sizes, the hybrid main memory size, the DRAM buffer size, and the types of regression prefetch engine, access latency, total hit rate, and energy consumption were carried out.

### 4.3.1. Optimal Size of Hybrid Main Memory

This section provides a comprehensive analysis and comparison of the performance of our proposed prefetching model with traditional model and other state-of-the-art prefetching models in the context of hybrid main memory. The main objective is to assess and evaluate the effectiveness and efficiency of our proposed prefetching methods and to demonstrate their superiority over existing models.

To obtain the optimal size of the mixed main memory, a scalability evaluation was performed. Four different sizes of DRAM (64, 128, 256, and 512 MB) and four different sizes of PCM (1, 2, 3, and 4 GB) were mixed and evaluated, but the size of the prefetch buffer was fixed as 15 MB.

Figure 7 displays the results of our evaluation regarding the execution times for different configurations of DRAM and PCM sizes. As the size of the PCM and ratio to DRAM buffer increases, the execution time decreases. Nonetheless, when the capacity of the PCM was changed from 1 to 2 GB, it showed the largest reduction in execution time compared to other hybrid main memory configurations. When holding the PCM capacity constant, the execution time exhibited a decrease with increasing DRAM size. However, as the capacity increases, the execution time reduction rate of the DRAM is significantly lower than that of the PCM. Therefore, according to the change in the PCM capacity, the execution time showed the greatest reduction rate.

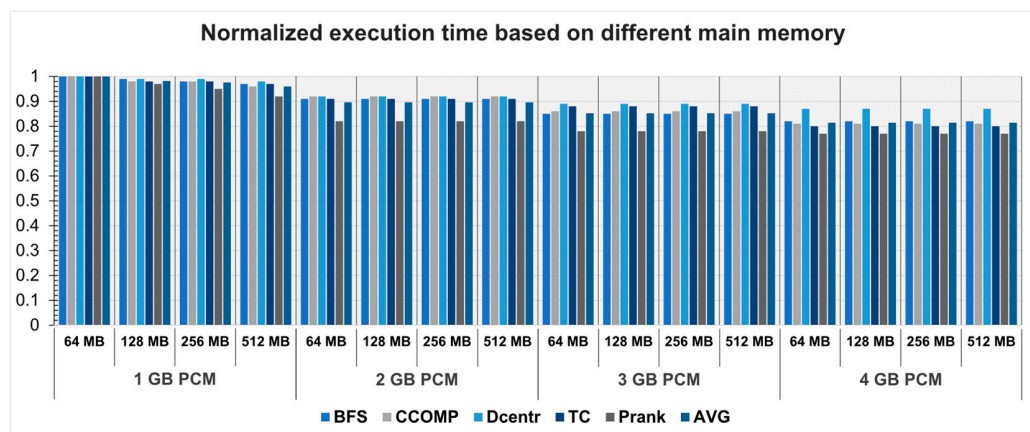

**Figure 7.** Execution time for different main memories.

The 2 GB PCM capacity of the recommended model with the largest reduction in execution time rate was selected. In the above experiment, considering the cost of DRAM (4×) compared with PCM and the ratio of DRAM size to PCM size, the 128 MB DRAM capacity of the recommended model was selected. Therefore, the capacity of the 128 MB DRAM and 2 GB PCM was finally selected as the optimal mixed main memory size for the performance evaluation of the proposed model.

### 4.3.2. Optimal Size of Dram Buffer

In this experimental study, our objective was to optimize the DRAM buffer size through an analysis of the changes in execution time and energy consumption for a range of DRAM buffer sizes. We configured buffer sizes of 2, 4, 8, 16, and 32 MB and plotted the overall execution time for the hybrid main memory system with different buffer sizes, as shown in Figure 8. We observed a steady decrease in execution time as the buffer size increased, on the average.

Based on our evaluations, we determined that a buffer size of 16 MB offers the ideal trade-off between system performance and cost efficiency for our proposed model with hybrid memory configurations. This was determined by analyzing the execution time for buffer sizes ranging from 2 to 32 MB, where the execution time steadily decreased as the buffer size increased, as shown in Figure 8.

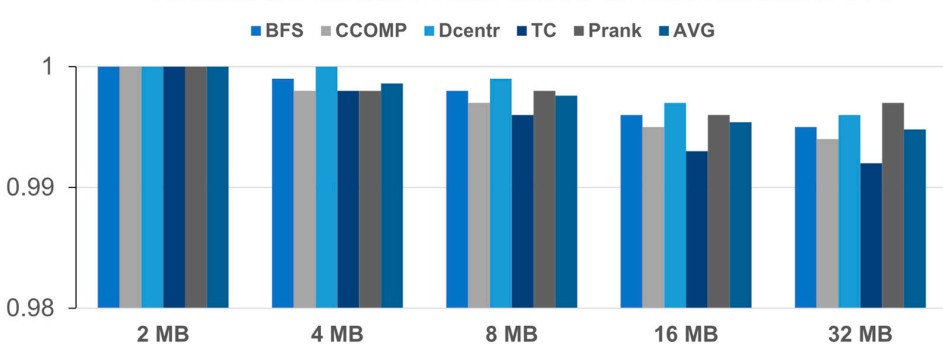

**Figure 8.** Execution time for different buffer sizes.

### 4.3.3. Optimal Size of Offset

To determine the optimal size for offset, we conducted an evaluation with various configurations to select the optimized capacity of DRAM buffer by measuring the enhancement in the execution time based on the varying offset sizes. As the memory mode can only be recognized when the offset is greater than three, the minimum representative of the offset chosen was five, with the offset sizes tested being for 5, 10, 15, 20, and 25.

Figure 9 presents the execution times for different offset sizes. As the size of the offset increases, the execution time decreases; however, when the offset size was 15, it exhibited the greatest reduction in execution time compared with other offset sizes, indicating that it was optimal.

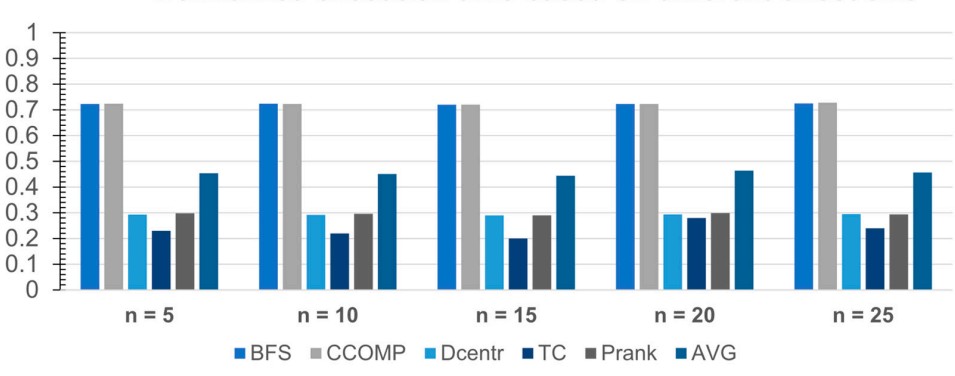

**Figure 9.** Execution time for different offset sizes.

### 4.3.4. Optimal Size of DRAM Buffer

This section presents a scalability evaluation that is conducted to obtain the optimal DRAM buffer size for the regression prefetcher of the hybrid main memory system. Three different regression preferences were evaluated. We conducted a comparative evaluation of various representative regression algorithms on traditional DRAM models to assess their performance. Specifically, we configured the proposed models in DRAM-based models, considering the cost and density aspects, as shown in Table 4.

As illustrated in Figure 10, we evaluated the total execution times of our benchmark applications using various regression prefetchers. The conventional (CONVEN) was tested without any prefetchers as the control group. It can be seen from the figure that, to a certain extent, neither the conventional LWR nor polynomial regression (Polyn.) can match the performance with the optimized adaptive algorithm we proposed. The test results show that our model has approximately 2–3% performance improvement compared with only other regression prefetching. Hence, by conducting an in-depth analysis of the execution time with respect to various regression algorithms, we were able to determine the optimized regression prefetcher with the optimal capacity.

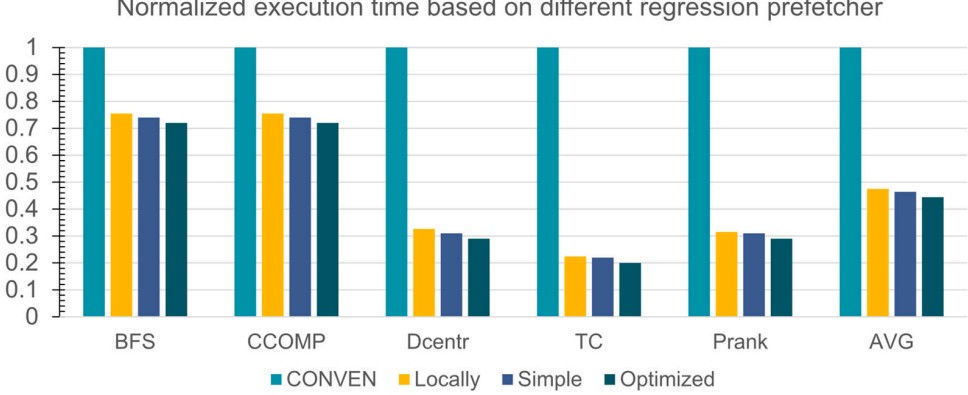

**Figure 10.** Execution time for different regression prefetchers.

### 4.4. Overall Performance Analysis

To validate our proposed model, we conducted a comparison between various prefetch methods, such as GHB, SMS, AMPM, Best-Offset, which are recognized as one of the state-of-the-art prefetchers, and Dynamic Recognition (DynamicR) [7,13,17]. In addition, we employed the LRU policy as the common cache replacement policy for all the models.

As shown in Figure 11, our proposed model outperformed all other compared models in execution times for all workloads that we have evaluated. Specifically, for the conventional DRAM model, our proposed model demonstrated a significant improvement of 56% in the execution time.

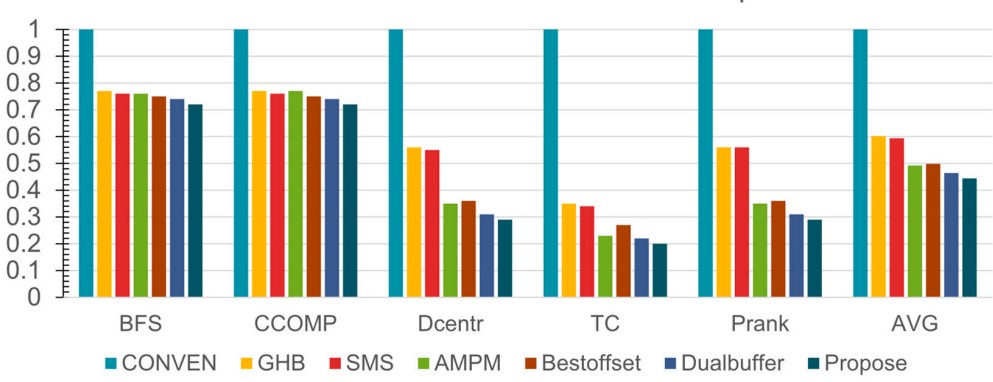

**Figure 11.** Execution time for graph processing.

Our proposed model outperformed the conventional prefetching models, achieving a 16% and 15% improvement in execution time over GHB and SMS, respectively. In addition, compared to AMPM and Best-Offset, the proposed model exhibited a 5% and 6% improvement in execution time, respectively. Notably, the proposed model demonstrated superior performance in execution time for both the degree-centrality (DC) and triangle-count (TC) workloads, which involve complex memory streams and intensive memory requests. Even with the advanced DynamicR prefetch engine, the performance improvement was about 3%.

Figure 12 presents a detailed comparison of energy consumption for various prefetcher-based hybrid main memory models, including our proposed one with various traditional prefetcher-based models. The experimental results indicate that our proposed model surpasses other comparable models through all workloads. The proposed model shows a 57% improvement in energy consumption compared to the conventional DRAM model. In comparison to existing prefetching models, the proposed model significantly surpasses their energy consumption performance, exhibiting a 16% and 15% better performance than GHB and SMS-based models, respectively. Furthermore, the proposed model out-

performs AMPM and Best-Offset prefetcher-based hybrid memory models, with 5% and 5.5% lower energy consumption, respectively. Moreover, the proposed model demonstrates superior energy efficiency compared to hybrid memory system models based on previous prefetchers, such as the AMPM and Best-Offset prefetchers, with a reduction in energy consumption of 5% and 5.5%, respectively. Overall, the proposed model showcases the best energy consumption performance among the evaluated models for the TC workload. Even when compared to the most advanced DynamicR prefetch engine, the proposed model demonstrates a 2% improvement in energy consumption, emphasizing its efficiency in terms of power consumption.

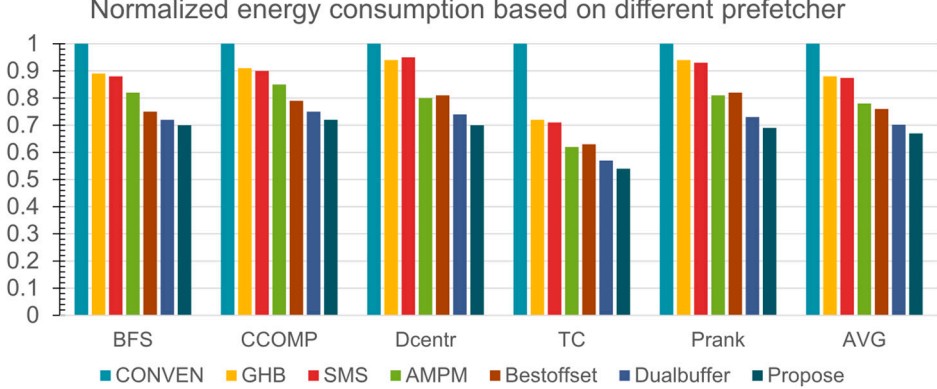

**Figure 12.** Energy consumption for graph processing.

Figure 13 compares the total hit rates of various prefetching methods, including our proposed model. The results show that our model outperformed conventional hybrid memory models based on various prefetchers, achieving an approximate 5% improvement compared to the conventional DRAM model. Our proposed model also achieved, on average, 2% improvement compared to other advanced models in terms of hit ratios. Notably, even the most advanced DynamicR prefetch engine only exhibited an improvement of approximately 0.3%, highlighting the significant effectiveness of our proposed model in improving hit rates.

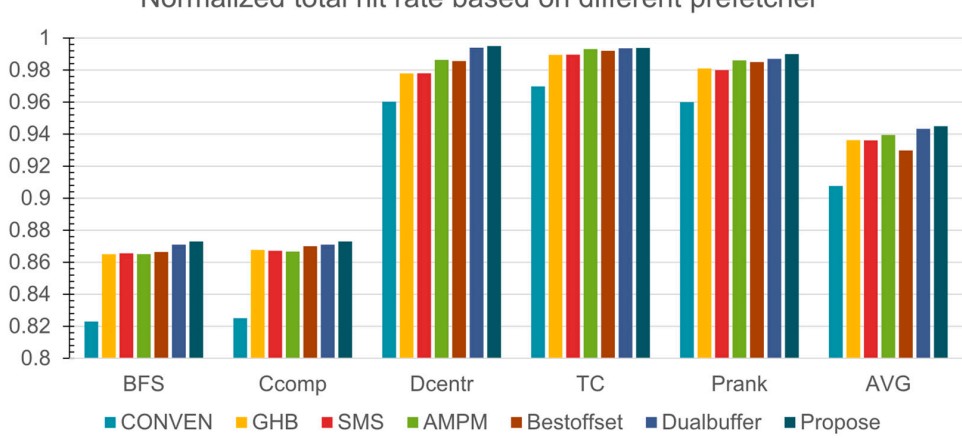

**Figure 13.** The total hit rate for graph processing.

The advanced point of our prefetch model had three primary reasons:

1. From a global perspective, pattern recognition was performed for different patterns, and three different prefetch engines were designed.
2. Local linear regression was optimized and improved, and an intelligent regression algorithm for memory prefetching in irregular patterns was proposed.

3.　　Several experiments were performed to choose the most suitable parameter configuration.

In conclusion, the proposed model demonstrated superior performance compared to previous models due to its intelligent linear regression prefetching schemes.

## 5. Conclusions

Our research introduces an optimized regression algorithm-based approach to enhance the hybrid main memory architecture, based on DRAM and PCM technologies specifically designed to accelerate graph-processing applications. By implementing a prefetching mechanism to hybrid memory systems, our proposed model efficiently places the required data into the DRAM buffer, resulting in improved average memory access latency for irregular memory requests from indirect and linked-list-based graph data structures. Conventional prefetching methods often struggle to generate appropriate prefetching candidates for the irregular memory access streams from graph processing workloads by detecting their patterns. To solve the irregularity of memory access patterns, next-lined, regular stride, and optimized linear regression prefetch engines were used to prefetch and switch appropriate prefetch engines. Based on our simulation-based evaluations, the proposed model utilizes the DRAM buffer-based DRAM-PCM hybrid main memory architecture and its optimized prefetching engine outperforms both conventional and other hybrid main memory models that are based on existing conventional prefetchers in terms of improved performance and energy efficiency.

The proposed scheme demonstrated a 56% and 57% better execution time and energy consumption, respectively, than the conventional DRAM model. Our proposed model surpasses other advanced models with a 12% improvement in execution time and a 9% improvement in energy consumption. Furthermore, it outperforms DynamicR by 3% in execution time and 2% in energy consumption. Moreover, it was also able to improve the hit rate by 2% on average. Thus, the proposed regression algorithm can improve the execution performance for graph-processing algorithms, incorporating key big-data processing characteristics.

**Author Contributions:** Conceptualization, M.Z. and S.-D.K.; methodology, M.Z., J.-G.K. and B.B.; software, M.Z. and Q.T.; formal analysis, M.Z. and Q.T.; writing—original draft preparation, M.Z.; writing—review and editing, J.-G.K. and S.-D.K.; supervision, S.-D.K. and B.B.; project administration, J.-G.K. and S.-D.K.; funding acquisition, S.-D.K., J.-G.K. and B.B. All authors have read and agreed to the published version of the manuscript.

**Funding:** This research was supported in part by the National Research Foundation of Korea (NRF) grant funded by the Korea government (MSIP) (No. 2022R1I1A1A0107213811), in part by Basic Science Research Program through the National Research Foundation of Korea (NRF) funded by the Ministry of Education (No. 2021R1I1A1A01059737), and in part by Samsung Electronics Co., Ltd (No. IO201209-07887-01).

**Institutional Review Board Statement:** Not applicable.

**Informed Consent Statement:** Not applicable.

**Data Availability Statement:** Data is contained within the article. The data presented in this study are available in Section 4.

**Conflicts of Interest:** The authors declare no conflict of interest.

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
