# Peer review of "Adaptive Regression Prefetching Algorithm by Using Big Data Application Characteristics"

_applsci, doi:10.3390/app13074436_

Round 1

Reviewer 1 Report

My comments are given below:

  Authors suggested an innovative prefetching algorithm with hybrid  memory structure based on the  DRAM and Phase-Change Memory.  They also presented the improved efficiency of hybrid memory for big data technologies. They also compared the numerical results, which they found to be quite impressive. So, I recommend this paper for publication in its current form.

Author Response

We would like to express our sincere appreciation on your precise indications in reviewing our paper. Your comments were very helpful in making our paper complete. Your positive remarks and recognition of the significance of my research are greatly appreciated.

Reviewer 2 Report

This article a prefetching algorithm was proposed for  graph-processing applications to prefetch the necessary data into the DRAM buffer and  reduce the latency of irregular memory accesses. The dual prefetching scheme is proposed. The proposed architecture consisting  of a DRAM-PCM hybrid main memory and DRAM buffer in total with an optimized engine performed better than the conventional and other existing prefetching models in  terms of both latency and energy efficiency. The scheme demonstrated a 56% and 57% better execution time and energy  consumption, respectively, than the conventional DRAM model. The proposed regression algorithm can improve the execution performance for graph-pro sensing algorithms, incorporating key big data processing characteristics.

The choice of the proposed methods is justified. The research is of practical importance.

 1. The sequence of operations of the arbitrator, as shown in Figure 3, is not clear. The referee should have the beginning and the end. The "Yes" actions of the branches are not complete.

2. The pictures 67 need to be improved.

Author Response

We would like to express our sincere appreciation on your precise indications in reviewing our paper. Your comments were very helpful in making our paper complete. Your positive remarks and recognition of the significance of my research are greatly appreciated. As you mentioned, Figure 3 is newly modified. And we also revised all the figures into scalable vector graphics for clarity.

Reviewer 3 Report

1- In the literature shall be adding a table as a comparison for related work.

2- Algorithm 1 and Algorithm 2 needed some details to be explained. 

3-In the Evaluations section, what is the time used? (second, millisecond,   microsecond, or nanosecond) shall be declared in the title of Figures 7 and 8.

4- Need to add to the number of references to be at least 30 up to dTE references.

5- the overall references shall be up to date and the older references need to remove.

Author Response

We would like to express our sincere appreciation on your precise indications in reviewing our paper. Your comments were very helpful in making our paper complete. Your positive remarks and recognition of the significance of my research are greatly appreciated.

[Q-1]: Thank you for your comment regarding adding a comparison table for related work in the literature review. While we appreciate your suggestion, we would like to respectfully note that the prefetching schemes discussed in the related works not only have differences but also exhibit mutual inheritance, making it challenging to concisely describe their features with only a few keywords in a table. As a result, we have opted to use clear and descriptive text to describe each scheme as thoroughly as possible.

We hope that this approach adequately conveys the information to the reader and appreciate your understanding in this matter.

[Q-2]: Thank you for your comment regarding the need for further explanation of Algorithm 1 and Algorithm 2. We would like to explain that these algorithms are intended to demonstrate the practical design of the major functionality described in their respective sections of the ext. They are intended to be illustrative major pseudocode, rather than fully detailed and executable code.

[Q-3]: Thank you for your comment regarding the declaration of the time unit used in Figures 7 and 8. We would like to explain that the data presented in the figures represent the normalized execution time, with the control group as the reference point for normalization. This control group is also the group with the longest execution time among all the experimental groups. Therefore, the data in the figures can be understood as the percentage of execution time relative to the control group.

In our simulation experiments, we used time parameters ranging from nanoseconds to milliseconds, as described in Table 4 in Section 4.3.4 of the manuscript.

[Q-4]: We appreciate your comments and suggestions to improve the quality of our research. Regarding the number of references, we have carefully considered your comments and would like to clarify that we have prioritized selecting relevant literature that directly relates to the scope of our research. However, we acknowledge that our study could benefit from a broader range of references. In the actual research process, we have extensively reviewed a wide range of research papers that are not listed in our manuscript. Therefore, we have taken into account all of the valuable insights and findings from these studies and have applied them to our research wherever applicable.

[Q-5]: As stated above, we have extensively reviewed and learned from a wide range of state-of-the-art research paper. Therefore, while increasing the number of references, we have also ensured that our references are up-to-date and removed outdated literature. Once again, we appreciate your thoughtful comments and suggestions, which have helped us to enhance the quality and relevance of our manuscript.

Reviewer 4 Report

1. Manuscripts are well written, clear and structured, so they are easy to understand.

2. The rationality described in the introduction is very good.

3. The research method is good, clear and rational and can be re-examined or continued.

4. The text can be accepted scientifically and empirically.

5. The research results have good novelty and state of the art.

6. Drawings/tables/schematics are appropriate, and made based on accurate data.

7. The author presents valid and reliable data.

8. The manuscript is easy to interpret and understand.

9. The data has been properly interpreted.

10. Conclusions have been presented in a manner consistent with the evidence and arguments presented.

11. The study has been written in a clear, comprehensive, and relevant to the field.

Author Response

(The authors gave the same response as above.)

Reviewer 5 Report

Please check the Title name, I think the first word " Application" should be deleted .

Author Response

We would like to express our sincere appreciation on your precise indications in reviewing our paper. Your comments were very helpful in making our paper complete. Your positive remarks and recognition of the significance of my research are greatly appreciated. We have considered your suggestion and agree that the first word "Application" seems to be of minor importance, so we have removed it from the title as suggested.